# ITEM RESPONSE SCALING LAWS: A MEASUREMENT THEORY APPROACH TO GENERALIZABLE NEURAL PERFORMANCE PREDICTION

## ABSTRACT

Classical neural scaling laws describe how the performance of large language models (LLMs) improves with increased compute, but they are typically estimated in aggregate across all questions in a benchmark, overlooking the information carried by individual questions. Item Response Theory (IRT) offers a principled way to address this by modeling per-question characteristics, though traditional IRT is limited to binary data with a Bernoulli loss. In pre-training downstream scaling, probabilities of producing the correct answer over the entire vocabulary yield more informative laws, while in test-time scaling, repeated sampling naturally gives rise to empirical probabilities. Empirical probability responses do not arise in human testing or LLM leaderboard evaluations, settings where IRT has shown success. To bridge this gap, we propose extending IRT with a Beta loss on empirical probability responses, naturally yielding Item Response Scaling Laws. We validate our framework in two large-scale studies: (1) pre-training downstream scaling, using 25 models from 6 families with up to 359 checkpoints on 15 NLP datasets; and (2) test-time scaling, using 15 models on 10 NLP datasets with up to 10,000 samples per question. In both cases, IRT-based approaches provide reliable and efficient estimates of scaling behavior while remaining interpretable and generalizable. [1]

## 1 INTRODUCTION

Modern large language models (LLMs) are general-purpose tools that offer diverse capabilities. Scaling laws, such as pre-training downstream scaling and test-time scaling, provide a principled framework for understanding and improving LLM performance. Pre-training downstream scaling laws characterize how a model's performance on downstream tasks improves as a function of the computational effort invested during pre-training, typically measured in floating-point operations (FLOP) (Biderman et al., 2023; Mishra et al., 2024). Test-time scaling laws describe how a model's inference performance on a benchmark (e.g., success rate) grows with the number of stochastic samples performed at test time (Brown et al., 2024; Hughes et al., 2024).

Deriving these scaling laws requires extensive data; however, practical studies are constrained to small experimental scales due to their costly nature. As a baseline, LLM evaluation leaderboards typically involve on the order of $10^6$ questions evaluated across $10^2$ models, amounting to $10^8$ total queries (Liang, 2023). Based on this, pretraining downstream scaling laws introduce around $10^2$ pretraining checkpoints per model, and test-time scaling laws introduce around $10^4$ samples per question. These additional dimensions inflate the total number of queries by several orders of magnitude. Consequently, existing studies often evaluate only a limited set of models and a subset of benchmark questions, and they rely primarily on mean score as the performance metric (Chen et al., 2024; Schaeffer et al., 2024; Brown et al., 2024).

The laws derived from limited experimental scales can exhibit surprising and unintuitive behaviors, including inconsistencies across different model–benchmark pairs. For example, Brown et al. (2024) derives a power-law test-time scaling relationship that, as Schaeffer et al. (2025) demonstrates, holds only for ill-structured response distributions in single-sample success rates. Such phenomena suggest that the benchmark-specific factors, such as the difficulty of each question, play a crucial role in shaping the scaling behavior. Moreover, mean score on a subset may not reliably reflect overall

---

[1]Code: `anonymous.4open.science/r/irsl-7560`

| | Definition | Classical Fitting Approach | IRT-based Fitting Approach |
|---|---|---|---|
| **Pre-training** Acc | $\mathbb{E}[\text{Acc}(i, \mathcal{D})] = \frac{1}{N} \sum_{j=1}^{N} p_{ij}$ | $a \cdot \sigma(b \cdot (\alpha \cdot \text{FLOP}_i^{-\beta} + \gamma - c)) + d$ | $\frac{1}{N} \sum_{j=1}^{N} \sigma(f(\text{FLOP}_i) - z_j)$ |
| **Pre-training** $p_{\text{Correct Choice}}$ | $p_{\text{Correct Choice}}(i, \mathcal{D}) = \frac{1}{N} \sum_{j=1}^{N} p_{\text{Correct Choice}}(i, j)$ | $g(\text{FLOP}_i)$ | $\frac{1}{N} \sum_{j=1}^{N} \sigma(f(\text{FLOP}_i) - z_j)$ |
| **Test-time** | $\text{pass@k}(i, \mathcal{D}) = \frac{1}{N} \sum_{j=1}^{N} 1 - (1 - p_{ij})^k$ | $\exp\left(-a \cdot k^{-b}\right)$ | $\frac{1}{N} \sum_{j=1}^{N} \left(1 - (1 - \sigma(\theta_i - z_j))^k\right)$ |

Table 1: The definitions, classical fitting approach, and IRT-based fitting approach for Acc, $p_{\text{Correct Choice}}$ (downstream pre-training scaling law), and pass@k (test-time scaling law). We use the Rasch model as a demonstration. Classical fitting approaches fit uninterpretable parameters specific to datasets and LLMs. Our IRT-based fitting approach learns question-level parameters (rather than dataset-level ones), while being interpretable and generalizable.

performance, nor does mean score on one dataset generalize to other datasets with similar measurement objectives but different difficulty levels (Truong et al., 2025). More fundamentally, these shortcomings arise because prevailing scaling-law formulations implicitly treat datasets as homogeneous inputs and focus exclusively on aggregate metrics, thereby obscuring the heterogeneity of item characteristics.

Item Response Theory (IRT), originating from psychology and educational testing, offers a principled way to address these limitations. IRT parameterizes question-level characteristics to model the probability that a test taker (human or LLM) answers a question correctly. It is typically applied in two phases: calibration and computerized adaptive testing (CAT). In the calibration phase, a binary response matrix is collected from test takers (human or LLMs) and used to estimate item characteristics such as question difficulty. With these calibrated parameters, CAT adaptively selects informative subsets of questions to reliably and efficiently evaluate new test takers. IRT has been highly successful in human testing, and recent work has shown its promise on binary response data from LLM leaderboards (Biderman et al., 2023; Mishra et al., 2024; Magnusson et al., 2025; Gadre et al., 2024). However, in both pre-training downstream scaling and test-time scaling laws, there is a need to model empirical probability responses rather than binary responses, an aspect absent from both human testing and LLM leaderboard evaluations. In the pre-training setting, mean scores often exhibit emergent behaviors and poor predictability (Lourie et al., 2025), whereas probabilities of producing the correct answer over the entire vocabulary preserve richer information and yield more consistent scaling laws (Schaeffer et al., 2024). In the test-time setting, repeated sampling naturally gives rise to empirical probabilities. Motivated by this, we propose extending IRT with a Beta loss to model empirical probabilities, yielding efficiently estimated, interpretable, and generalizable Item Response Scaling Laws. Our contributions are:

- We conduct a large-scale study on 25 models from 6 model families, ranging from 111M to 13B parameters, with up to 359 pretraining checkpoints, on 15 popular natural language datasets, to demonstrate the superiority of our item response pretraining downstream scaling laws: interpretable, generalizable, and giving robust estimates of the scaling behavior with limited budgets.

- We conduct a large-scale study on 15 models across 10 popular natural language datasets with up to 10,000 repeated sampling steps. We demonstrate that IRT gives more reliable estimates of the scaling behavior in an efficient manner, surpassing traditional approaches, while being interpretable and generalizable.

By embedding the scaling law within the IRT framework, our approach provides a theoretically principled and empirically validated alternative to conventional aggregate performance scaling.

## 2 RELATED WORK

**Pre-training Downstream Scaling Law** Many neural networks exhibit power-law scaling of the pre-training loss as a function of the amount of compute, data, or parameters used for training (Hestness et al., 2017; Kaplan et al., 2020; Bahri et al., 2021; Hernandez et al., 2021; Hoffmann et al., 2022; Muennighoff et al., 2024). Unlike predicting loss, predicting downstream performance from scale is generally harder (Lourie et al., 2025; Schaeffer et al., 2024). However, recent work has demonstrated it can be done based on a two-step prediction that chains together predictions from

scale to loss and loss to downstream performance (Biderman et al., 2023; Mishra et al., 2024; Magnusson et al., 2025; Gadre et al., 2024).

**Test-time Scaling Law**  Test-time scaling laws characterize how a model's performance on a benchmark (e.g., success rate) improves as the number of stochastic samples drawn at inference increases (Brown et al., 2024; Hughes et al., 2024). Schaeffer et al. (2025) demonstrate that such laws hold only for ill-structured response distributions in single-sample success rates. Kang et al. (2025) develop scalable test-time inference strategies that balance sample efficiency and performance gains under limited budgets. Dorner et al. (2025) establish that the shape of a verifier's ROC curve governs the effectiveness of rerolling and best-of-N, formalizing limits of test-time scaling with imperfect verifiers. Zhang et al. (2025) survey the test-time scaling literature and outline open challenges for reliability and generalization.

**Measurement Theory-based AI evaluation**  Several recent works adopt Item Response Theory (IRT) as a foundation for AI evaluation using binary responses and Bernoulli loss (Truong et al., 2025; Hofmann et al., 2025; Kipnis et al., 2025; Madaan et al., 2024; Polo et al., 2024). Polo et al. (2025) introduce Sloth scaling laws, which model benchmark performance through low-dimensional latent skills influenced by compute, offering a measurement-theoretic perspective that improves prediction and interpretability across LLM families. Heineman et al. (2025) propose a framework based on signal and noise to quantify benchmark reliability, showing that higher signal-to-noise ratios correlate with more accurate model ranking and lower scaling law prediction error, and introducing interventions to improve evaluation robustness.

## 3 METHOD

Item Response Theory (IRT) provides an elegant and parsimonious mathematical framework to model the natural interaction of LLMs and benchmark questions. We show how, under this framework, various known scaling laws arise naturally and how the framework facilitates efficient, interpretable, and generalizable scaling evaluation. We show the definitions, classical fitting approaches, and IRT-based fitting approaches of the scaling laws in Table 1.

### 3.1 ITEM RESPONSE THEORY

Item Response Theory refers to a class of probabilistic latent variable models that explain the relationship between the test taker's latent ability, the question's characteristics (e.g., difficulty), and the observed response from the test taker to the questions (Baker, 2001; Van der Linden et al., 2000). A central model in IRT is the Rasch model (Rasch, 1993), where each test-taker (here, an LLM) has an ability parameter $\theta$, and each question has a difficulty parameter $z$. A higher $\theta$ denotes greater ability, while a higher $z$ signifies a more difficult question. Let $y$ denote the response of the test taker to the question, where $y = 1$ if the response is correct and 0 otherwise. The probability of a correct response is modeled by $p(y = 1 \mid \theta, z) = \sigma(\theta - z)$, where $\sigma$ is the sigmoid function. Another widely adopted model in IRT is the 2PL model (Lord, 1952; Birnbaum, 1968), which adds a discrimination parameter $a$ to capture how sharply a question differentiates between test-takers of different abilities, modeling the probability of a correct response as $p(y = 1 \mid \theta, z, a) = \sigma(a(\theta - z))$. In this case, A higher $a$ indicates that the question is more effective at distinguishing between abilities close to its difficulty level. The difficulty $z$ and the discrimination $a$ are collectively referred to as the item parameters. The use of IRT consists of two phases: calibration, which estimates item parameters, and computerized adaptive testing (CAT), which enables efficient evaluation for new test-takers.

In traditional human testing, the most common approach is to perform calibration on binary response data, where each question is scored as either correct or incorrect. A binary response matrix $Y \in \{0,1\}^{M \times N}$ is collected, where $M$ and $N$ denote the number of test takers and questions, respectively. Each entry $Y_{ij}$ represents a response of test taker $i$ to question $j$. With the binary response matrix, the item parameters can be estimated via either MLE or EM by minimizing the Bernoulli loss between the IRT-predicted probabilities and the observed binary responses $\mathcal{L}_{\text{Bernoulli}} = -\sum_{i=1}^{M} \sum_{j=1}^{N} [Y_{ij} \log p_{ij} + (1 - Y_{ij}) \log(1 - p_{ij})]$ (Bock & Aitkin, 1981; Chalmers, 2012; Wu et al., 2020).

Unlike human testing, LLM evaluation provides empirical response probabilities, which convey richer information than binary responses. In the pre-training downstream scenario, the probability of producing the correct answer among the entire vocabulary, $p^{\text{Vocab}}(\text{Correct Choice})$, as well as among the set of multiple-choice options, $p^{\text{Choices}}(\text{Correct Choice})$, provides a richer signal than accuracy of

binary responses (Schaeffer et al., 2024). In the test-time scaling scenario, each question is queried thousands of times with independent samples to estimate pass@1, defined as the probability that a single independent sample answers the question correctly. To exploit those probability information for calibration, we collect the probability matrix $P \in \mathbb{R}^{M \times N}, 0 \leq P_{ij} \leq 1$ and estimate the item parameters by minimizing the Beta loss between the IRT-predicted probabilities and the observed empirical probabilities $\mathcal{L}_{\text{Beta}} = -\sum_{i=1}^{M} \sum_{j=1}^{N} \log \left( \text{BetaPDF}(P_{ij}; \mu_{ij}, \phi) \right)$, where $\mu_{ij} = p_{ij}$ is the IRT-predicted probability, and $\phi$ is a precision (inverse variance) parameter. Traditional calibration on binary data usually demands a large and diverse set of LLMs, which is costly to obtain. By leveraging empirical response probabilities, reliable calibration can be achieved with fewer test-takers, thereby significantly lowering the computational cost of querying LLMs.

Given a question bank $\mathcal{Q}$ with calibrated item parameters, computerized adaptive testing can efficiently evaluate new test-takers using fewer questions. At query step $t$, given the current estimate of the test taker's ability $\theta_{\text{new}}^t$ and the calibrated item parameters, the next question $q_j^{*t}$ is selected as the most informative question, guided by an acquisition function such as Fisher information $\mathbb{I}$ (Meijer & Nering, 1999; Chang, 2015; Magis et al., 2017). This process iterates between the question selection step $q_j^{*t} = \arg\max_{q_j \in \mathcal{Q}^t} \mathbb{I}$   $\mathcal{Q}^{t+1} = \mathcal{Q}^t \setminus \{q_j^{*t}\}$ and the ability estimation step $\theta_{\text{new}}^{t+1} = \arg\max_{\theta_{\text{new}}^t} \sum_{j=1}^{t} \log p_{\text{new},j}$. Ability estimation can also be carried out using binary responses with a Bernoulli loss or empirical probability responses with a Beta loss.

## 3.2 Pre-training Downstream Item Response Scaling Law

The pre-training downstream scaling law characterizes how model performance on a benchmark scales with the compute used during model pre-training. Performance of model $i$ on dataset $\mathcal{D}$ (e.g., MMLU) with $N$ questions indexed by $j$ is usually quantified by the expected mean accuracy, which is often approximated with an empirical mean accuracy:

$$\mathbb{E}[\text{Acc}(i, \mathcal{D})] = \mathbb{E}\left[ \frac{1}{N} \sum_{j=1}^{N} Y_{ij} \right] = \frac{1}{N} \sum_{j=1}^{N} \mathbb{E}[Y_{ij}] = \frac{1}{N} \sum_{j=1}^{N} p_{ij}.$$

$$\mathbb{E}[\text{Acc}(i, \mathcal{D})] \approx \text{Acc}(i, \mathcal{D}) = \frac{1}{N} \sum_{j=1}^{N} Y_{ij}, \text{ where } Y_{ij} \sim \text{Bern}(p_{ij}).$$

(1)

The binary variable $Y_{ij}$ is model $i$'s response to the $j$-th question (1 for a correct response and 0 otherwise). Classical linear approximation model Acc as a function of the model $i$'s pretraining compute FLOP$_i$ (Bhagia et al., 2024; et al., 2024; Gadre et al., 2024).

$$\text{Acc}(i, \mathcal{D}) \approx a \cdot \sigma(b \cdot (L_i - c)) + d \text{ where } L_i \approx \alpha \cdot \text{FLOP}_i^{-\beta} + \gamma,$$

(2)

where $\sigma$ is the sigmoid function, $a, b, c$, and $d$ are dataset-specific parameters for $\mathcal{D}$, $L_i$ is the LLM-specific cross-entropy loss, and $\alpha, \beta, \gamma$ are LLM-specific parameters for model $i$. None of these scaling parameters is interpretable.

We propose to model the probability of test taker $i$ correctly answering question $j$ from benchmark $\mathcal{D}$ with IRT. For example, under the Rasch model, this probability is given as $\hat{p}_{ij} = \sigma(\theta_i - z_j)$:

$$\mathbb{E}[\text{Acc}(i, \mathcal{D})] \approx \frac{1}{N} \sum_{j=1}^{N} \hat{p}_{ij} = \frac{1}{N} \sum_{j=1}^{N} \sigma(\theta_i - z_j) \quad \theta_i = f(\text{FLOP}_i),$$

(3)

where $\theta_i$ generalizes across questions that measure the same construct, and $z_j$ generalizes across different LLMs. $\theta_i$ can be interpreted as the ability of test taker $i$, $z_j$ can be interpreted as difficulty of question $j$. Modeling pre-training downstream scaling law from Acc gives emergent behavior and low predictability (Lourie et al., 2025; Schaeffer et al., 2024). There is growing interest in modeling the law from the probability of producing the correct answer among the entire vocabulary, $p^{\text{Vocab}}(\text{Correct Choice})$, which preserves more information (Schaeffer et al., 2024):

$$p_{\text{Correct Choice}}(i, \mathcal{D}) = \frac{1}{N} \sum_{j=1}^{N} p_{\text{Correct Choice}}(i, j).$$

(4)

Traditional approaches tend to fit a regression model with LLM-specific and dataset-specific parameters $g$: $p_{\text{Correct Choice}}(i, \mathcal{D}) \approx g(\text{FLOP}_i)$. We propose to use IRT to model $p_{\text{Correct Choice}(i,j)}$. For example, under the Rasch model, this probability is given as $\widehat{p_{\text{Correct Choice}(i,j)}} = \sigma(\theta_i - z_j)$:

$$p_{\text{Correct Choice}}(i, \mathcal{D}) \approx \frac{1}{N} \sum_{j=1}^{N} \sigma(\theta_i - z_j), \quad \theta_i = f(\text{FLOP}_i). \tag{5}$$

### 3.3 Test-time Item Response Scaling Law

Test-time scaling investigates the predictable relationship between a model's success rates on a benchmark and the number of independent samples generated during inference (Brown et al., 2024; Hughes et al., 2024). The model $i$'s success rate on dataset $\mathcal{D}$ with $N$ questions indexed by $j$ and $H$ samples indexed by $k$ is quantified by $\text{pass@k}(i, \mathcal{D})$:

$$\text{pass@k}(i, \mathcal{D}) = \frac{1}{N} \sum_{j=1}^{N} \text{pass@k}(i, j) = \frac{1}{N} \sum_{j=1}^{N} 1 - (1 - \text{pass@1}(i, j))^k \tag{6}$$

where the $\text{pass@k}(i, j) = 1$ if at least one of the $k$ generated responses of model $i$ to $q_j$ is correct, and $\text{pass@1}(i, j)$ denote the probability that a single independent sample of model $i$ correctly answers question $q_j$ (Schaeffer et al., 2025). Brown et al. (2024) model the negative log success rate falls as a power law with the number of independent samples $k$: $\text{pass@k}(i, \mathcal{D}) \approx \exp(-ak^{-b})$, where $a, b > 0$ are uninterpretable model-specific and benchmark-specific parameters. We propose to model $\text{pass@1}(i, j)$ using IRT. For example, under the Rasch model, this probability is given by $\text{pass@1}(i, j) = p_{ij} = \sigma(\theta_i - z_j)$. Instead of fitting LLM-specific and benchmark-specific constants $a, b$, we models $\text{pass@k}(i, \mathcal{D})$ as a function of IRT parameters:

$$\text{pass@k}(i, \mathcal{D}) \approx \frac{1}{N} \sum_{j=1}^{N} \left(1 - (1 - \hat{p}_{ij})^k\right) = \frac{1}{N} \sum_{j=1}^{N} \left(1 - (1 - \sigma(\theta_i - z_j))^k\right), \tag{7}$$

where $\theta_i$ generalizes across questions that measure the same construct, and $z_j$ generalizes across different LLMs, both are interpretable.

## 4 Experiments

We demonstrate the advantages of the downstream item response scaling law in Section 4.1 and of the test-time item response scaling law in Section 4.2.

### 4.1 Pre-training Downstream Item Response Scaling Law

**Experimental Setup**  We collect two sets of responses across LLMs and datasets. (1) The first consists of the binary responses from 13 models spanning 3 model families with up to 359 checkpoints, evaluated on 13 datasets covering both exact-match and multiple-choice tasks. This yields a binary response matrix $Y_1^{\text{pretrain}}$ The data are queried using the HELM framework at temperature 0 (Liang, 2023). Truong et al. (2025) calibrated question difficulties on these datasets using the Rasch model with binary responses from 42–91 diverse LLMs on the HELM leaderboard. Under the same query setup, we reuse these calibrated difficulties in our experiments. (2) The second is obtained from Schaeffer et al. (2024) and contains outputs from 5 model families (see Schaeffer et al. (2024) for details). For our experiments, we select three out of the 12 multiple-choice datasets in Schaeffer et al. (2024). Each output includes a binary response $y$, a probability response $p^{\text{Vocab}}(\text{Correct Choice})$, and a probability response $p^{\text{Choices}}(\text{Correct Choice})$, resulting in one binary response matrix $Y_2^{\text{pretrain}}$ and two probability response matrices $P_2^{\text{pretrain pvocab}}$ and $P_2^{\text{pretrain pchoices}}$. The data are queried using the LM Evaluation Harness at temperature 0 (Gao et al., 2023). Hofmann et al. (2025) calibrate item parameters on these datasets using the 2PL model with binary responses from 102 diverse LLMs on the Open LLM Leaderboard (Beeching et al., 2023). Under the same query setup, we reuse these calibrated item parameters in our experiments.

**Binary-IRT CAT and Scaling Law Fit**  We first apply IRT with a Bernoulli loss (denoted Binary-IRT) to the binary response matrix $Y_1^{\text{pretrain}}$. Using the calibrated question difficulties, we conduct CAT on the LLM checkpoints with a question budget of 100. We then compare the $\theta$–FLOP curve

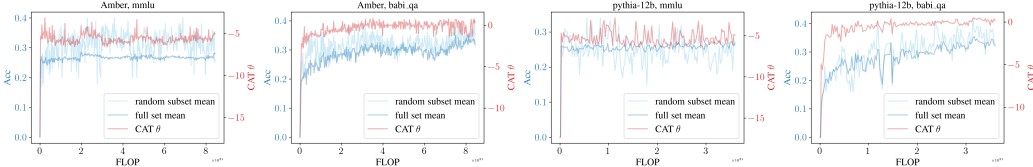

Figure 1: Comparison of binary-IRT (red) and Acc curves (light blue) during pre-training on a subset, across 2 LLMs and 2 datasets. The full-set Acc (dark blue) serves as ground truth. The blue lines use the left $y$-axis (Acc), while the red line use the right $y$-axis (CAT $\theta$). Subset binary-IRT yields lower variance than subset Acc. But both IRT and ground truth Acc show emergent behavior and exhibit poor predictability.

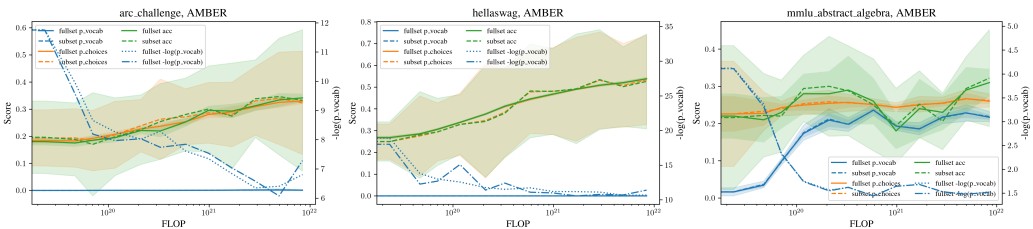

Figure 2: Naive scaling behavior of three metrics (Acc, $p^{\text{Vocab}}$(Correct Choice), $p^{\text{Choices}}$(Correct Choice)) versus FLOP of AMBER on 3 datasets. Because $p^{\text{Vocab}}$(Correct Choice) is often too small to visualize, we also plot its negative log probability. The left $y$-axis corresponds to the three metrics, and the right $y$-axis corresponds to $-\log p^{\text{Vocab}}$. Shaded regions indicate $\pm 3$ standard deviations estimated from subset sampling or bootstrap. $p^{\text{Vocab}}$(Correct Choice) or its negative log probability does not show emergent behaviors and exhibits strong predictability.

obtained from CAT with the Acc–FLOP curve computed on a random subset of 100 questions. The full-set Acc serves as the ground truth. As shown in Figure 1, IRT yields lower variance and closely tracks the ground-truth curve. To assess evaluation stability, we report the total variation (TV) following Hofmann et al. (2025). For each model $i$ and benchmark $\mathcal{D}$ with $V$ checkpoints, let $x_i^v(\mathcal{D})$ denote the measured performance (either Acc or $\theta$) at checkpoint $v$. The normalized total variation is defined as $\text{TV}(i, \mathcal{D}) = \frac{V}{V-1} \cdot \frac{\sum_{v=1}^{V-1} \left| x_i^{v+1}(\mathcal{D}) - x_i^v(\mathcal{D}) \right|}{\left| x_i^V(\mathcal{D}) - x_i^1(\mathcal{D}) \right|}$, where lower values indicate lower step-to-step variance and thus higher evaluation quality. Averaged across all datasets and models, subset Acc yields TV $= 39.66$, while subset IRT achieves TV $= 26.72$, demonstrating its superior stability. However, neither IRT nor ground truth Acc addresses the challenging questions of downstream scaling laws: emergence and predictability.

**Naive Scaling Behavior of $p_{\text{Correct Choice}}$**   Schaeffer et al. (2024) find that $p^{\text{Vocab}}$(Correct Choice) retains substantially more information than binary responses, offering a potential avenue to address emergence and predictability. Using $Y_2^{\text{pretrain}}$, $P_2^{\text{pretrain pvocab}}$, and $P_2^{\text{pretrain pchoices}}$, we examine the naive scaling behavior of three metrics: Acc, $p^{\text{Vocab}}$(Correct Choice), and $p^{\text{Choices}}$(Correct Choice). Figure 2 plots these metrics against FLOP of AMBER on three datasets, averaged across the full set and on a random subset of 50 questions. We observe that $p^{\text{Vocab}}$(Correct Choice) (or its negative log) exhibits clear scaling behavior. While $p^{\text{Vocab}}$(Correct Choice) is too small to be effectively modeled by IRT on ARC Challenge and HellaSwag, it appears feasible on MMLU. Accordingly, we apply Beta-IRT to both $p^{\text{Vocab}}$(Correct Choice) and $p^{\text{Choices}}$(Correct Choice) for MMLU. Moreover, the average $p^{\text{Vocab}}$(Correct Choice) across the 50-question subset of MMLU closely aligns with the full-set curves with low variance, suggesting that naive probability-based metrics already provide strong approximations that may be difficult for IRT to improve upon, as we describe next.

**Beta-IRT CAT and Scaling Law Fit**   We apply IRT with a Beta loss (denoted Beta-IRT) to the probability response matrix $P_2^{\text{pretrain pvocab}}$ and $P_2^{\text{pretrain pchoices}}$. Using the calibrated item parameters, we conduct CAT on the LLM checkpoints with a question budget of 50. In Figure 3, we use the average $p^{\text{Vocab}}$(Correct Choice) on the full set as the ground truth. We compare the average $p_{\text{Correct Choice}}$ on the subset, and estimate $\theta$ using Beta-IRT CAT on probability responses with the budget. We

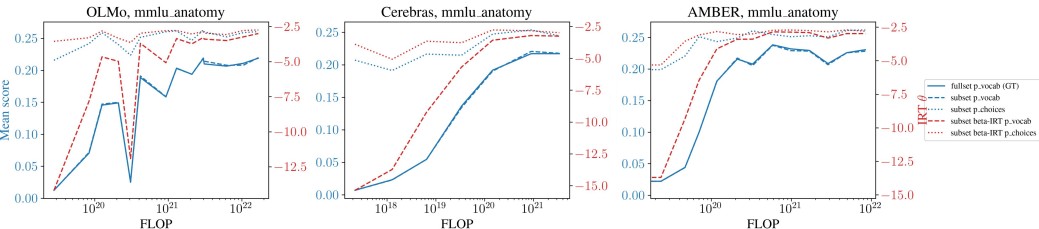

Figure 3: Scaling behavior of three model families on MMLU Anatomy. The blue lines use the left $y$-axis (Mean score), while the red lines use the right $y$-axis (IRT $\theta$). Beta-IRT effectively captures $p_{\text{Correct Choice}}$, which is more predictable and less emergent.

observe that both the $\theta$ estimated from Beta-IRT CAT and the average $p_{\text{Correct Choice}}$ exhibit smooth increasing trends that closely align with the ground truth. This suggests that Beta-IRT effectively captures $p_{\text{Correct Choice}}$, which is more predictable and less emergent, making it a suitable candidate for modeling downstream scaling laws. We note, however, that while subset Beta-IRT does not surpass the subset average of $p_{\text{Correct Choice}}$ in estimation accuracy or stability, it performs comparably and remains valuable for its interpretability and generalizability, as we describe next.

**IRT Ability Estimates Generalize across Questions Measuring the same Construct**   We use the scaling performance of Pythia-12B on WikiFact (from $Y_1^{\text{pretrain}}$) as a demonstration. The scaling law is characterized by the relationship between Acc and the training FLOP of the 154 Pythia-12B checkpoints. We split the questions into an easy subset for training and a difficult subset for testing according to the calibrated difficulties. For the IRT model, we first estimate the model's ability parameter $\theta$ on the training set using pre-calibrated difficulties; we then model $\theta$ as a function of FLOP via Kernel Ridge Regression: $\theta = f_{\text{KRR}}(\text{FLOP})$. Finally, we infer the test set accuracy by combining $f_{\text{KRR}}(\text{FLOP})$ with the pre-calibrated test set difficulties. As shown in Figure 4, the IRT model closely matches the ground truth curve on both the training and test sets. This shows that $\theta$ generalizes across questions with different difficulties but measuring the same construct, while the traditional scaling law remains test set-specific. We replicate this experiment across 13 benchmarks and 13 models from $Y_1^{\text{pretrain}}$. Figure 7 presents the difference in test mean squared error (MSE) with ground truth between the classical scaling law (which fits on the train ground truth and stays the same for the test set) and the item response scaling law (which generalizes to the

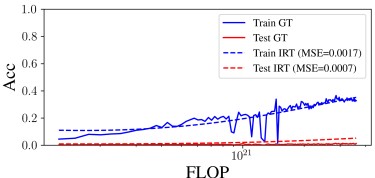

Figure 4: Pythia-12B checkpoints on WikiFact. Acc versus FLOP on the easy (train, blue) and difficult (test, red) question splits. Solid: GT scaling curves; dotted: IRT curves. The $\theta$ estimated from the easy question set generalizes to the hard question set, accurately predicting the scaling behavior on the hard question set, while the traditional approach fits dataset-specific scaling parameters.

test set). From the consistent positive MSE difference, we conclude that the item response scaling law demonstrates superior generalization across questions measuring the same construct.

This is a core property of IRT known in psychology as test-set invariance, meaning that ability estimates remain robust across subsets of varying difficulty. This feature is particularly important in the pre-training setting. For example, in an industrial setting where training and evaluation are handled by separate teams, increasing model ability with more compute causes CAT to adaptively choose more difficult questions for evaluation (Hofmann et al., 2025). By doing so, CAT ensures that different subsets are presented to the model, which can reduce contamination and lead to more reliable evaluation.

### 4.2   TEST-TIME ITEM RESPONSE SCALING LAW

**Experimental Setup**   We collect two probability matrices, denoted as $P_1^{\text{testtime}}$ and $P_2^{\text{testtime}}$. The matrix $P_1^{\text{testtime}}$ contains the responses of 8 LLMs on 552 questions drawn from 6 multiple-choice benchmarks. For each entry, the empirical $\text{pass}@1$ is estimated by averaging over $10,000$ samples, with queries performed under a few-shot setting and temperature 1. The matrix $P_2^{\text{testtime}}$ contains the responses of 12 latest LLMs on 120 questions from 4 latest benchmarks, including both exact

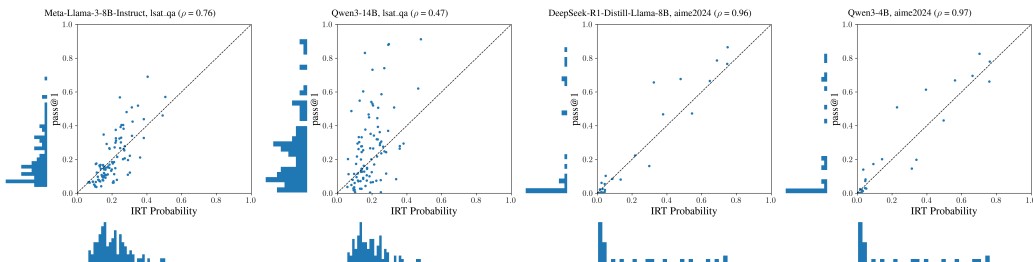

Figure 5: Consistency between (1) IRT-predicted probability with $\theta$ estimated from 50 or 30 questions and 50 samples ($x$-axis), and (2) pass@1 estimated from 10,000 or 2,500 samples ($y$-axis). Each point represents a question in the benchmark. From left to right, the four subplots are training LLM in $P_1^{\text{testtime}}$, test LLM in $P_1^{\text{testtime}}$, training LLM in $P_2^{\text{testtime}}$, and test LLM in $P_2^{\text{testtime}}$. IRT-predicted probabilities correlate closely with pass@1.

match and multiple-choice tasks. Here, the empirical pass@1 is estimated by averaging over $2,500$ samples, with queries performed without few-shot examples but using chain-of-thought prompting at temperature $0.6$.

**Beta-IRT Calibration** We adopt the Rasch model for both matrices, leaving out 4 LLMs for CAT. Question difficulty parameters $z$ are calibrated on the remaining LLMs using the Beta loss. We report two correlation metrics for calibration performance: the Spearman correlation between the estimated difficulty $z$ and the average empirical pass@1 of each question across the LLMs used for calibration, denoted as $\rho_{\text{train}}$; and the Spearman correlation between $z$ and the average empirical pass@1 across the LLMs left out for CAT, denoted as $\rho_{\text{test}}$. For $P_1^{\text{testtime}}$, we obtain $\rho_{\text{train}} = 0.97$ and $\rho_{\text{test}} = 0.80$. For $P_2^{\text{testtime}}$, we obtain $\rho_{\text{train}} = 0.99$ and $\rho_{\text{test}} = 0.95$.

Although leaderboard-calibrated difficulties are publicly available (Hofmann et al., 2025; Truong et al., 2025), we recalibrate question difficulties on the test-time probability matrices, because test-time scaling is typically conducted with a positive temperature to enable diverse sampling, whereas leaderboard evaluations are usually performed at temperature 0 for reproducibility. Our analysis shows that difficulties obtained under different evaluation setups can sometimes exhibit low correlation. For $P_1^{\text{testtime}}$, the questions overlap with those used in Truong et al. (2025). In Truong et al. (2025), question difficulties are calibrated from a binary response matrix of 42–91 test takers collected from the HELM leaderboard (Liang, 2023). We compare the estimated difficulties between our setting and theirs: across the six benchmarks, three exhibit high correlation while the other three do not, likely due to differences in evaluation setup (Figure 8). These results suggest there is hope to transfer difficulty estimates calibrated from leaderboards to test-time scaling scenarios, but they are not always reliable.

**Beta-IRT CAT** With calibrated question difficulties, we perform CAT using the Beta loss on both the LLMs included in calibration and those held out, with a sample budget of 50. For $P_1^{\text{testtime}}$, we set the question budget to 50, and for $P_2^{\text{testtime}}$, we set it to 30. Given the estimated LLM ability $\theta$, we compute the correlation between the IRT-predicted probability $\sigma(\theta - z)$ and the empirical pass@1 estimated from all available samples. Figure 5 presents four subplots, corresponding to one training (included in calibration) and one test (held out) LLM on each of $P_1^{\text{testtime}}$ and $P_2^{\text{testtime}}$. We report the Spearman correlation averaged across LLMs and datasets between IRT-predicted probability and pass@1: $\rho_{\text{train}}^{P1} = 0.70, \rho_{\text{test}}^{P1} = 0.53, \rho_{\text{train}}^{P2} = 0.92, \rho_{\text{test}}^{P2} = 0.88$. We observe that the IRT-predicted probabilities correlate closely with pass@1. However, when pass@1 is exactly zero, the IRT model fails to produce an exact zero probability, which introduces discrepancies in the law fitting process, as we describe next.

**Scaling Law Fit** We report 3 law curves on pass@k or $-log$ pass@k versus number of samples $k$ in Figure 9: (1) the full unbiased (GT) curve serves as the ground truth. In this case, pass@k is estimated from all available samples (10,000 or 2,500 samples) and all available questions using the unbiased and numerically stable estimator of Chen et al. (2021): $\widehat{\text{pass@k}}(i,j) = 1 - \binom{H-c_{ij}}{k}/\binom{H}{k}$, where $H$ is the total number of available samples, $c_{ij}$ is the number of correct samples of LLM $i$ on question $q_j$ among those samples. (2) In the sub pass@1 curve, pass@k is estimated from subset questions (50 or 30 questions) and subset samples (50 samples): $\widehat{\text{pass@k}}(i,j) = \mathbb{E}_j[1 - (1 -$

$\hat{p}_{ij})^k]$, where $\hat{p}_{ij} = c_j/H$. (3) In the sub beta-IRT curve, $\hat{z}$ is estimated from calibration, and $\hat{\theta}$ is estimated from subset questions (50 or 30 questions) and subset samples (50 samples). pass@k is then estimated as $\widehat{\text{pass@k}}(i, j) = \mathbb{E}_j[1 - (1 - \hat{p}_{ij})^k]$, where $\hat{p}_{ij} = \sigma(\hat{\theta} - \hat{z})$. In the left half of Figure 9, we observe that the IRT law curves tend to achieve pass@k $= 1$ while the ground truth curves do not. This arises because IRT probabilities never reach exact zero: both $\theta$ and $z$ are typically modeled as approximately normal, and even a large negative difference such as $\theta - z = -6$ yields $\sigma(-6) \approx 0.0025$. This behavior can be interpreted as an inherent limitation of IRT; one possible extension would be to introduce a prior that induces a binning effect. We also note that the ground-truth curves may be resolution-limited due to finite samples: if an LLM never answers a question correctly in the limited samples, the ground truth estimates become exactly zero.

In psychometrics, questions that are never correctly answered are typically discarded because they provide no discriminatory power (Lord, 1980). Following this principle, we filter out questions with pass@1 $\leq 0.01$ and show empirically that this improves predictive reliability of IRT. After filtering, we obtain the right half of Figure 9, where the sub beta-IRT curve provides a closer estimate than the sub pass@1 curve under a limited query budget. We report the mean absolute errors (MAEs) of $-\log$ pass@k between the subset estimates and the ground-truth estimates for both IRT and pass@1. To highlight the relative advantage, we compute $\text{MAE}_{\text{pass@1}} - \text{MAE}_{\text{beta-IRT}}$ and visualize the results in Figure 10, where positive values (red) indicate superiority of IRT. Across nearly all LLMs and datasets, IRT achieves more reliable estimates of test-time scaling laws in an efficient manner, while being interpretable and generalizable.

**IRT Ability Estimates Generalize across Questions Measuring the same Construct** We use the scaling performance of Qwen3-4B on Global MMLU Lite as a demonstration. We split the questions into an easy subset for training and a difficult subset for testing according to the calibrated difficulties. For the IRT, we first estimate the model's ability parameter $\theta$ on the training set using pre-calibrated difficulties; we then combine $\theta$ with the pre-calibrated difficulties of the test set to predict the probability of correct responses. As shown in Figure 6 (left), the IRT curves closely match the ground truth curves on both the training and test sets. This shows that $\theta$ generalizes across questions with different difficulties but measuring the same construct, while the traditional scaling law remains test set-specific. In Figure 6 (right), we visualize $\widehat{\text{pass@1}}$ as a function of question difficulty $z$. The LLM ability $\theta$ estimated generalizes effectively to the test set, producing a scaling curve that accurately predicts pass@1 on more difficult questions.

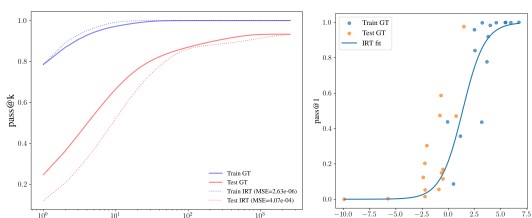

Figure 6: Qwen3-4B on Global MMLU Lite. **Left:** pass@k versus $k$ on the easy (train, blue) and difficult (test, red) question splits. Solid: GT scaling curves; dotted: IRT curves. **Right:** pass@1 as a function of question difficulty $z$. The LLM ability $\theta$ estimated from the easy set generalizes effectively to the hard set, accurately predicting pass@1 and giving accurate estimation for pass@k scaling on more difficult questions, while the traditional approach remains dataset-specific.

## 5 DISCUSSION, LIMITATIONS, AND FUTURE WORK

In the scaling laws literature, pre-training downstream curves are typically fit only to final checkpoints; by contrast, we fit to intermediate checkpoints of pretrained models. Our estimation of FLOP may be inaccurate because learning signals are nonuniform over training (e.g. due to warmup and decay schedules). Future work includes introducing a prior into IRT that induces a binning effect in test-time settings, fitting a shared latent ability that generalizes across benchmarks under the hypothesis that most benchmarks measure the ability to autoregress text (Truong et al., 2025; Kipnis et al., 2025), exploring alternative probabilistic models beyond IRT, extending to other scaling laws (e.g., observational scaling law (Ruan et al., 2024), pre-training validation-loss scaling law (Kaplan et al., 2020), in-context scaling laws (Arora et al., 2025)), and extending IRT to non-binary assessments (Ostini & Nering, 2006).

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

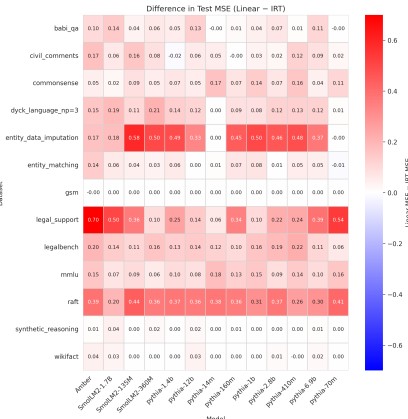

Figure 7: Heatmap of the difference in test MSE (Classic – IRT) across 13 benchmarks (rows) and 13 LLMs (columns) in $Y_1^{\text{pretrain}}$. Consistent positive (red) values indicate that the classical scaling law incurs higher error than the item response scaling law on the test set, demonstrating the latter's superior generalization across question subsets with the same measurement objective but differing in difficulty.

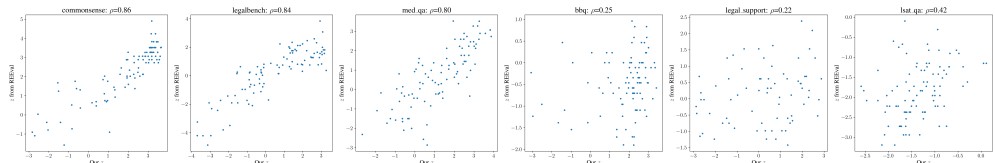

Figure 8: Consistency of question difficulty estimates between our calibration ($x$-axis) and Truong et al. (2025) ($y$-axis). Each point represents a question. Across the six benchmarks, three exhibit high correlation while the other three do not. There is hope to transfer difficulty estimates calibrated from leaderboards (temperature = 0) to test-time scaling scenarios (temperature > 0), but they are not always reliable.

## A    THE USE OF LARGE LANGUAGE MODELS (LLMS)

We used large language models (LLMs) as general-purpose assistive tools for two purposes: (1) polishing the writing of the paper, and (2) providing coding assistance. In all cases, the outputs generated by LLMs were carefully reviewed, verified, and modified by the authors before inclusion. The authors take full responsibility for all content presented in this paper.

## B    ADDITIONAL FIGURES

### B.1    PRE-TRAINING DOWNSTREAM ITEM RESPONSE SCALING LAW

### B.2    TEST-TIME ITEM RESPONSE SCALING LAW

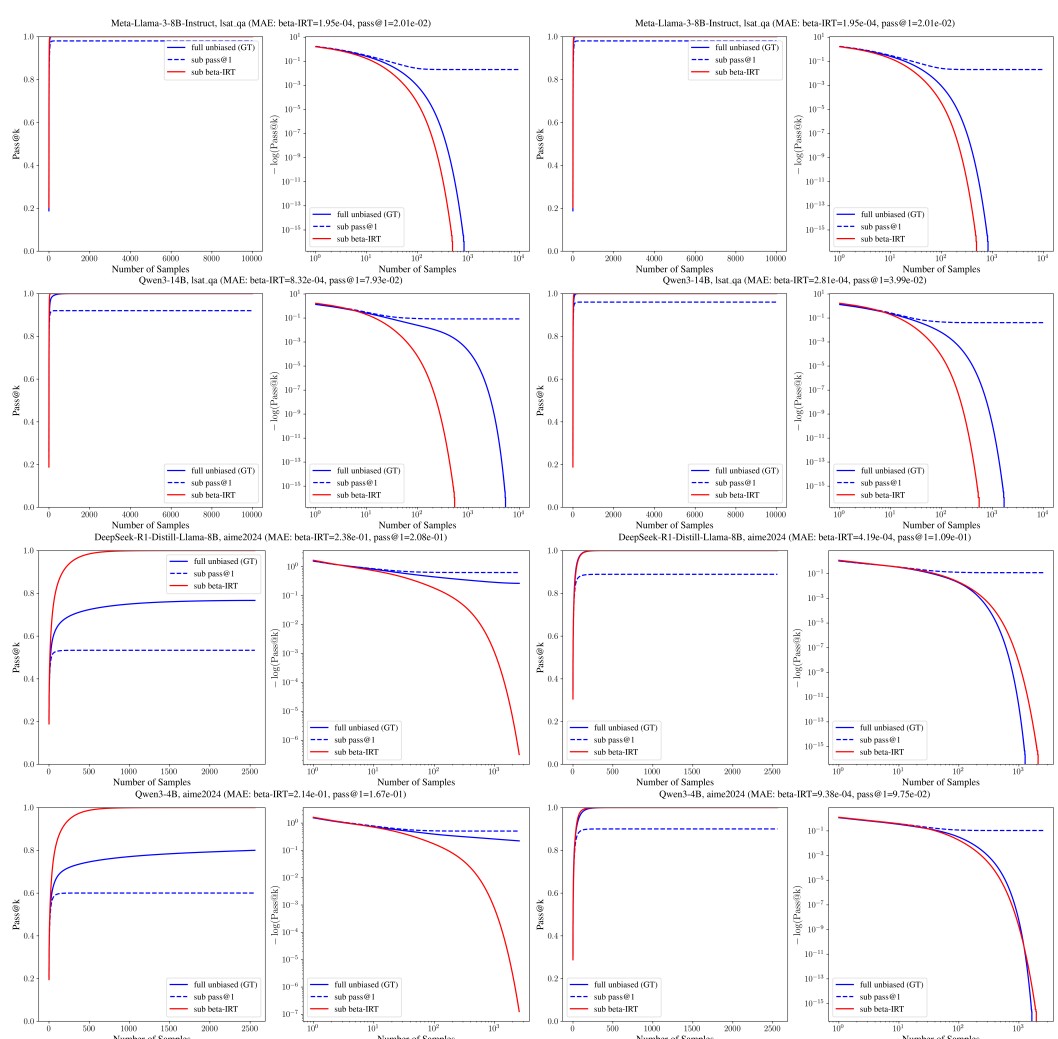

Figure 9: Left 2 columns: pass@k or $-log$ pass@k versus number of samples $k$ before filtering. Right 2 columns: pass@k or $-log$ pass@k versus number of samples $k$ after filtering. Row 1 to 4: a training LLM in $P_1^{\text{testtime}}$, a test LLM in $P_1^{\text{testtime}}$, a training LLM in $P_2^{\text{testtime}}$, and a test LLM in $P_2^{\text{testtime}}$. The MAEs of $-log$ pass@k between estimates on subset questions and subset samples and GT are reported in subplot titles. After filtering out questions with low pass@1, IRT achieves more reliable estimates of test-time scaling laws than traditional approaches with limited budget.

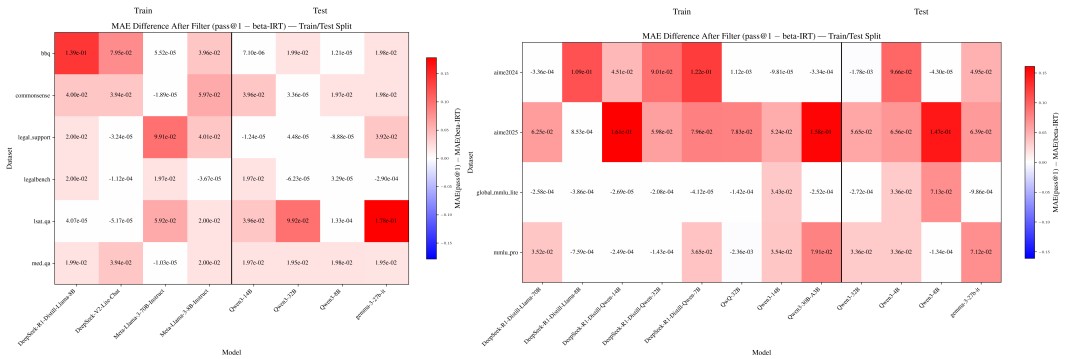

Figure 10: $\text{MAE}_{\text{pass@1}} - \text{MAE}_{\text{beta-IRT}}$ across LLMs and datasets. Left: $P_1^{\text{testtime}}$, Right: $P_2^{\text{testtime}}$. Across nearly all LLMs and datasets, IRT achieves more reliable estimates of test-time scaling laws in an efficient manner.

