# OpenReview forum: "Item Response Scaling Laws: A Measurement Theory Approach to Generalizable Neural Performance Prediction"
_ICLR.cc/2026/Conference — Submitted to ICLR 2026_

### Official Review · Reviewer_dyjt · 2025-10-23

**Soundness:** 2
**Presentation:** 1
**Contribution:** 2
**Rating:** 2
**Confidence:** 3

**Summary:**

This paper introduces Item Response Scaling Law, an Item Response Theory (IRT)-based method for generalized performance prediction of neural networks. The proposed method evaluates models on a small set of samples (easy ones) to estimate the model parameters, such as LLM ability and question difficulty. Then, it predicts model performance on the second set of samples (hard ones). Experimental results demonstrate IRT's potential to serve as an interpretable and generalizable tool for predicting model performance.

**Strengths:**

1. The idea of using IRT is interesting and may have high potential in model performance prediction.
2. The main idea is clear.

**Weaknesses:**

1. This paper is hard to follow. The section and paraphrase arrangement can be improved:

    (1) Contents in Section 3.1, where you introduce IRT, and contents in Section 2, where you introduce the prediction formulation of test-time scaling, should move to Section 2.

    (2) The 2nd and 3rd paragraphs of Section1 are a bit ambiguous. I assume the 2nd is about experimental scale; the 3rd is too poorly structured, so I don't understand the main argument. I suggest you rewrite the two sections by explicitly listing down ((1), (2),...) challenges. In the 4th paragraph, you can thus make it clearer on why IRT addresses them (e.g., we address issue (1) by ...; (2) by...).

2. Given that your method builds on the Rasch model, it will be clearer to introduce its formulation in Section 2. You may also argue why other IRT models, like 3PL, were not chosen.

3. Line 177 seems to lack "," between two equations, and the subscript "new" in lines 177-178 is redundant.

4. In line 213, you write p^Vocab (Correct Choice); but from line 215 it becomes p_{Correct Choice}.

5. In lines 255 and 263, I don't recommend using subscripts "1" and "2" to split between two sets of responses. "1" and "2" do not convey information and do not clearly differentiate between the two sets.

6. In Figure 3, the y-axis label overlaps with the legend.

7. I don't recommend explaining Figures in the appendix (Figures 7~9) as the main paper's analysis. The main paper should be self-contained. Reviewers, though encouraged, are not obliged to review the appendix.

**Questions:**

1. [a] measures the difficulty of each question (later splitting them into different question groups based on difficulty) using the Brier score instead of IRT. I believe IRT has a more fine-grained depiction of difficulty, and thus it might be worth discussing as a related work.

2. In the 2nd paragraph of Section 1, you mention that previous work struggles at the experimental scale—How does IRT address this issue? My understanding is that IRT addresses the issue of granularity instead of scale.

3. In Figure 2's subset-based prediction, how did you do subset sampling and plot the final prediction line? If I didn't misunderstand, the reason why the prediction line matches the full set's curve is simply because the sample mean is an unbiased estimator. For example, if you have 10 samples [a, b,..., i, j] and at each time you uniformly sample 3 of them to calculate the average. Doing this 100 times leads to a number close to the average of [a, b,..., i, j].

4. I am confused about Figures 1 and 3. Why is the right y-axis IRT theta? Isn't it a fixed estimated value? How to compare it with the left axis (Mean score)? Left axis is the Mean score of what?

5. What are baseline methods for Beta-IRT to compare with?

a. [U-shaped and Inverted-U Scaling behind Emergent Abilities of Large Language Models](https://openreview.net/forum?id=jjfve2gIXe)

---

### Official Review · Reviewer_rLDS · 2025-11-01

**Soundness:** 3
**Presentation:** 2
**Contribution:** 3
**Rating:** 6
**Confidence:** 3

**Summary:**

This paper addresses a critical limitation of classical neural scaling laws for large language models (LLMs): their reliance on aggregate metrics (e.g., mean accuracy) across benchmark questions, which overlooks individual question characteristics and leads to poor generalizability and interpretability. To solve this, the authors propose Item Response Scaling Laws (IRSL), an extension of Item Response Theory (IRT) from psychology/educational testing to model LLM performance scaling.

**Strengths:**

IRSL establishes a new paradigm for scaling laws—rooted in measurement theory—instead of empirical curve-fitting. This provides a theoretical basis for generalizability, which classical laws lack.

**Weaknesses:**

The Beta loss uses a precision parameter $\(\phi\)$, but the paper does not explain how $\(\phi\)$ is selected (e.g., cross-validation, fixed value). Sensitivity to $\(\phi\)$ is untested, which raises questions about whether $\(\phi\)$ choices drive results.

**Questions:**

1. You filtered questions with pass@1 ≤0.01 to mitigate IRT’s zero-probability limitation. Why 0.01 specifically? Did you test other thresholds (e.g., 0.005, 0.02) and, if so, how did they affect MAE and the number of retained questions? Is there a theoretical basis (e.g., IRT discriminatory power) for choosing this threshold?
2. How did you select the value of $\(\phi\)$ for the Beta loss? Did you perform a sensitivity analysis to test how different $\(\phi\)$ (e.g., 0.01, 1, 100) impact calibration accuracy (Spearman $\(\rho\)$) and scaling law fit (MAE)? If $\(\phi\)$ is optimized via cross-validation, what was the validation procedure (e.g., leave-one-dataset-out)?

---

### Official Review · Reviewer_uUtS · 2025-11-01

**Soundness:** 3
**Presentation:** 2
**Contribution:** 2
**Rating:** 4
**Confidence:** 2

**Summary:**

The authors propose Item Response Theory (IRT) scaling law, which models per-question difficulty and per-LLM ability, to predict downstream performance. It conducts two main experiments, one on pre-training downstream scaling and another on test-time scaling. The authors conclude from both experiments that IRT scaling laws are a useful framework to predict downstream task performance.

**Strengths:**

The motivation and theory part is written well. The introduction of Item Response Theory to scaling law is natural and novel. I agree that modelling per-question characteristics is an interesting next-step that deepens our understanding of the scaling law.

**Weaknesses:**

W1: The organization and presentation of the experiment part can be strengthen. Upon reading it multiple times, I feel it is challenging for me (and potentially other readers as well) to grasp (1) does the method work well, (2) how does it compare to and whether it improves upon existing methods. I feel many experiments lack a clear take-away and I find it challenging to grasp what is the main finding/result for every experiment.

W2. The authors should strengthen the presentation of evidence regarding  effectiveness of their method. In several experiments, the authors conclude effectiveness from comparing IRT's curve and the ground truth and saying there is a good match, for example Figure 1, Figure 4, and Figure 5. But this seems subjective and readers might not agree (for example, there are mismatches in Figure 4 and 5). Could the authors present more objective criteria for effectiveness of their method?

W3. It would be very helpful if the authors could include more baseline comparisons. In most experiments, the author direct compare their method to ground truth and with some level of match, the authors argue their method predicts well. It lacks a situational context for readers to interpret such level of match. Presenting more baseline comparisons would help solve W2 as well.

Direction for improvement: I feel it would strengthen the paper greatly if the authors could go beyond analysis and demonstrate downstream application of IRT scaling laws as well. Does it give new insights on scaling LLMs and could the authors test it?

**Questions:**

1. Line 298: The authors said "As shown in Figure 1, IRT [...] closely tracks the ground-truth curve". I could not understand how it is supported from Figure 1. binary-IRT (red) reports CAT \theta and full set Acc is the ground-truth. These two are two different metrics. How to support the finding that "IRTlosely tracks the ground-truth curve"? I would encourage the authors to clarify in paper as well.

2. In line 268 ~ 307 and Figure 1, could the authors compare to more baselines, such as the Classical linear approximation model, Equation 2 in paper. As the "random subset mean" baseline does not leverage any information.

3. I wonder why in Figure 1, experiments are conducted in a much narrower FLOP range, not on a logarithm scale such as in Figure 2?

4. Figure 3 has a similar problem, that while the authors said "Beta-IRT effectively captures pCorrect Choice, which is more predictable and less emergent", Beta-IRT and average pCorrect Choice report two different metrics and I am not sure how the figure supports this conclusion.

5. From Figure 3, it appears that subset p_vocab is the mostly close with the fullest p_vocab, which is the ground truth. Can I interpret that as subset p_vocab gives better productivity than the proposed subset beta-IRT p_vocab?

6. I have some questions over the selection of models & datasets, which the authors are encouraged to justify. For example, model sizes are much narrower in Figure 1; a specific subset of MMLU, MMLU anatomy, is used in Figure 3;

7. I think Figure 1 & Figure 4 are better plotted as a scatter plot rather than as a pure line plot, as the most important aspect is the points representing models.

8. The train-test split in the paragraph from Line 343 to 365 is decided by the calibrated difficulties, which themselves are a factor in the IRT model. I have concerns that this split might be problematic. Could the authors justify or consider using an external factor that split the dataset into an easy and difficult part?

9. I encourage the authors to include the classic scaling law as a baseline comparison in the paragraph from line 343 to 365, and Figure 4, to provide evidence that IRT is superior.

10. In Figure 4, isn't that the Test IRT became systematically higher than Test GT, which seems contradicting line 354 "As shown in Figure 4, the IRT model closely matches the ground truth curve on both the training and test sets"? The authors are encouraged to discuss this.

11. I encourage the authors to include other methods and compare in Figure 5 experiment.

12. I'm less sure about how to interpret the Spearman correlation reported in line 421, and whether "IRT-predicted probabilities correlate closely with pass@1" (line 422). Visually in Figure 5, it is also questionable to me IRT probabilities predict pass@1 well. While I understand that scaling law might inherently be a challenging task to predict, the authors should present more context & baselines to aid readers in understanding their method's improvement and superiority. The experiment shown in Figure 6 has the same problem.

---

### Official Review · Reviewer_nMND · 2025-11-03

**Soundness:** 4
**Presentation:** 3
**Contribution:** 3
**Rating:** 6
**Confidence:** 2

**Summary:**

This paper proposes Item Response Scaling Laws (IRSL), a framework that recasts neural scaling laws within the formalism of Item Response Theory (IRT) from psychometrics.
Instead of modeling dataset-level performance (e.g., accuracy vs. FLOPs), the authors treat each model as a “test taker” and each benchmark question as an “item” with latent difficulty $z_j$.
Model ability \theta_i and item difficulty $z_j$ are jointly modeled through a Rasch or Beta-IRT formulation, where $p(y_{ij}=1) = \sigma(\theta_i - z_j)$.
This allows the authors to interpret scaling behavior in terms of latent ability estimation, leading to smoother, more stable scaling curves and improved sample efficiency via Computerized Adaptive Testing (CAT).
Extensive experiments across pre-training and test-time scaling settings show that IRSL provides smoother, lower-variance trends compared to traditional aggregate fitting, and the latent ability \theta generalizes across question subsets.

**Strengths:**

- The work introduces a theoretically principled bridge between scaling laws and measurement theory.
Framing scaling as the interaction between model ability and item difficulty is conceptually elegant and potentially influential for future interpretability-driven scaling research.
- The paper evaluates both pre-training downstream scaling and test-time scaling, covering multiple model families, FLOP ranges, and benchmarks. The results (e.g., Figs. 1–4, 6–10) consistently demonstrate that IRT-based estimation yields smoother curves and reduced variance (e.g., 33% TV reduction in Fig. 1).
- The latent parameters \theta and z provide an interpretable decomposition of model performance that could, in principle, generalize across benchmarks.

**Weaknesses:**

1. Based on my understanding, the framework critically relies on calibrated item difficulties $z_j$, either imported from prior leaderboards (e.g., HELM, LM-Eval) or newly estimated using model response data through Beta loss.
This reliance means IRSL cannot operate in a truly predictive regime: for a new benchmark without prior model responses, the difficulty parameters are unidentifiable, and thus the scaling curve cannot be extrapolated.
In other words, IRSL functions as a measurement and efficiency framework rather than a predictive scaling law.
The paper’s claim of “generalizable prediction” holds only when item-level calibration is already available, not when facing unseen tasks, which is a key difference from conventional scaling laws that aim for zero-shot extrapolation.
Empirical discussion of cross-benchmark transfer (Fig. 8) supports this concern, showing that calibrated difficulties are not stable across decoding setups.
2. Modeling the empirical probabilities with a Beta likelihood implicitly assumes unimodal, symmetric response uncertainty.
In practice, LLM confidence distributions can be heavy-tailed or multi-modal due to sampling temperature, decoding randomness, or calibration bias.
The robustness of this assumption is not analyzed.

**Questions:**

See weaknesses

---

### Meta-Review · Area_Chair_He3T · 2025-12-30

**Summary:**

The paper proposes Item Response Scaling Laws (IRSL), which integrate Item Response Theory (IRT) from psychometrics into the study of neural scaling laws. Rather than focusing on aggregate benchmark accuracy, the authors model per-item difficulty and per-model ability using a novel Beta loss formulation to handle empirical probabilities.

While the reviewers generally praised the theoretical elegance and the novelty of bridging measurement theory with scaling laws, there are several weaknesses and concerns unresolved:
- Reviewers questioned whether the framework is truly predictive or merely an efficient measurement tool for existing benchmarks
- There are several important technical details regarding the Beta loss hyperparameters and data filtering thresholds were omitted
- The presentation of results was found to be confusing, with several reviewers noting a lack of objective baselines and subjective interpretations of good fits in the plots

**Reviewer Concerns:**

As the authors did not participate in the rebuttal phase or provide a response to the reviewers, no concerns were addressed

**Reviewer Scores:**

Since the authors did not respond and attend the rebuttal, reviewer scores won't increase and even have probabilities to decrease.

---

### Decision · Program_Chairs · 2026-01-26

Reject